# Hesperitin-Copper(II) Complex Regulates the NLRP3 Pathway and Attenuates Hyperuricemia and Renal Inflammation

**DOI:** 10.3390/foods13040591

**Published:** 2024-02-15

**Authors:** Xi Peng, Kai Liu, Xing Hu, Deming Gong, Guowen Zhang

**Affiliations:** 1State Key Laboratory of Food Science and Resources, Nanchang University, Nanchang 330047, China; pxjxja@126.com (X.P.); ncuspyliukai@163.com (K.L.); hx0726@ncu.edu.cn (X.H.); dgong01@gmail.com (D.G.); 2Department of Biological Engineering, Jiangxi Biotech Vocational College, Nanchang 330200, China

**Keywords:** hesperitin-Cu(II) complex, hyperuricemia, oxidative stress, renal protection

## Abstract

Background: Hyperuricaemia (HUA) is a disorder of purine metabolism in the body. We previously synthesized a hesperitin (Hsp)-Cu(II) complex and found that the complex possessed strong uric acid (UA)-reducing activity in vitro. In this study we further explored the complex’s UA-lowering and nephroprotective effects in vivo. Methods: A mouse with HUA was used to investigate the complex’s hypouricemic and nephroprotective effects via biochemical analysis, RT-PCR, and Western blot. Results: Hsp-Cu(II) complex markedly decreased the serum UA level and restored kidney tissue damage to normal in HUA mice. Meanwhile, the complex inhibited liver adenosine deaminase (ADA) and xanthine oxidase (XO) activities to reduce UA synthesis and modulated the protein expression of urate transporters to promote UA excretion. Hsp-Cu(II) treatment significantly suppressed oxidative stress and inflammatory in the kidney, reduced the contents of cytokines and inhibited the activation of the nucleotide-binding oligomerization domain (NOD)-like receptor thermal protein domain associated protein 3 (NLRP3) inflammatory pathway. Conclusions: Hsp-Cu(II) complex reduced serum UA and protected kidneys from renal inflammatory damage and oxidative stress by modulating the NLRP3 pathway. Hsp-Cu(II) complex may be a promising dietary supplement or nutraceutical for the therapy of hyperuricemia.

## 1. Introduction

A fasting blood uric acid level above 420 μmol/L in men and 360 μmol/L in women on a normal purine diet is known as hyperuricemia (HUA) [1]. HUA has no obvious symptoms in the early stage, but chronically high levels of serum uric acid may cause the formation of urate crystals in various organs, resulting in a series of complications [2,3,4]. At present, there are approximately 120 million patients with HUA in China, of whom about 95% are male; it affects younger patients at an increasing rate. HUA has become one of the most life-threatening diseases in the world [5].

HUA is caused by disorders of purine metabolism and UA excretion. The liver xanthine oxidase (XO) and adenosine deaminase (ADA) are involved in purine metabolism that catalyzes hypoxanthine and adenine into UA and O^2^ [6]. Serum UA cannot be broken down into more soluble allantoin; it can only be excreted, as blood is transported to the kidneys and large intestine [7,8]. In the process of UA reabsorption, urate transporter 1 (URAT1) transports UA to the renal tubular epithelium cells, and glucose transporter 9 (GLUT9) subsequently transports UA to the blood to complete the reabsorption of UA [9,10]. When blood flows through the S2 segment of the renal tubule, about 5% UA will be exchanged with α-ketoglutarate under the action of organic anion transporter 1 (OAT1), transported into the renal tubule, and finally excreted out of the body [11]. Therefore, the abnormal increases in XO and ADA activity in the liver and the impairment of renal excretion function will lead to the accumulation of UA to form HUA [12]. Urate crystallization causes necrosis of renal tubular epithelial cells and leukocytes and activates the NLRP3 inflammatory pathway. In addition, urate crystallization results in an imbalance of redox homeostasis, and reactive oxygen species (ROS) that are not cleared by superoxide dismutase (SOD) and catalase (CAT) in time will attack mitochondria in podocytes, activating the NLRP3 inflammatory pathway [13,14]. The NLRP3 increases the expression of inflammatory cytokines, which induces a series of inflammatory reactions [15,16,17]. In the treatment of HUA, the XO inhibitors were used to inhibit UA synthesis, and benzbromarone was used to promote UA excretion [18]. However, these drugs often cause adverse reactions in clinical use, such as fever, allergy, abdominal pain, diarrhea, liver, and kidney injuries [19], which makes their applications limited. 

Flavonoids are a class of active ingredients in plants whose wide range of pharmacological activities and few toxic side effects make them an important resource for functional foods and new drug development [20,21]. Chen et al. [22] reported that baicalein lowered UA levels in mice by suppressing the activity of XO. Zhu et al. [23] found that kaempferol and quercetin in the extract of cyclocarya paliurus alleviated HUA and reduced renal inflammation in mice. Hesperidin, the main active ingredient of citrus fruits of the Rutaceae family, has diuretic, antiviral, antibacterial, and stomach pain relief activities [24,25]. Hsp possessed significant anticancer potential by mediating apoptosis of cancer cells and hypouricemic ability in HUA rats by inhibiting XO activity [26]. Haidari et al. [27] found that Hsp reduced the serum UA level of HUA rats by suppressing liver XO activity. However, with similar properties to most flavonoids, Hsp has some shortcomings (e.g., insufficient solubility, diminished stability, poor bioavailability), which limit its development as a UA-lowering food function factor. Therefore, taking Hsp as the lead compound, to modify its structure in order to develop new UA-lowering food functional factors or drugs with higher activity, better water solubility, stability, and bioavailability, has important application value.

A large number of studies have indicated that flavonoid compounds form complexes with metal ions, significantly improving their physicochemical properties and biological activity [28]. For example, the synthesized kaempferol-3-neohesperidin-VO (IV) complex showed good hypoglycemic activity [29], the luteolin-Yb(III) complex exhibited a strong anti-inflammatory effect [30], and the antioxidant activity of Cu, Fe, and Mg complexes of quercetin was stronger than that of ligand quercetin [31]. Cu(II) could bind to Chrysin to form a metal complex that showed stronger inhibitory activity on XO than that of the ligand chrysin [32]. Based on the abundant resources of Hsp and an edible trace element of copper that is essential to the human body, we synthesized and characterized Hsp-Cu(II) complex, and found that the complex was non-toxic and strongly inhibited XO with ten times and nearly twice the strengths of Hsp and the clinical drug allopurinol (Alp) [33], respectively, indicating that the complex may have a strong UA-lowering effect, and may be a potential new type of XO inhibitor. However, to our knowledge, whether the complex has a hypouricemic effect in vivo is not known.

Therefore, a mouse diseased of HUA was conducted to study the Hsp-Cu(II) complex’s impact on the serum uric acid, blood urea nitrogen and creatinine level, cytokines, renal injury, and activities of key enzymes (XO and ADA) in the liver and SOD and CAT in the kidneys. The regulation of the complex on the expression of related renal UA transport proteins (including URAT1, GLUT9, and OAT1) and the NLRP3 inflammatory pathway in the mice, and the hypouricemic and nephroprotective effects and mechanisms of the complex were then explored. This investigation can provide a rationale and laboratory guidance for the study and exploitation of Hsp-Cu(II) complexes as food supplements or nutraceuticals with UA-lowering effects.

## 2. Materials and Methods

### 2.1. Chemicals and Reagents

Hsp (purity > 97%), carboxymethyl cellulose sodium (CMC-Na), hypoxanthine, Alp, and paraformaldehyde were all bought from Aladdin Chemistry Co. (Shanghai, China). Oteracil potassium was acquired from Macklin Biochemical Technology Co., Ltd. (Shanghai, China). The Hsp-Cu(II) complex was synthesized and characterized in our laboratory [33]. The biochemical assay kits of UA (C012-2-1), blood urea nitrogen (BUN, C013-2-1), creatinine (Cr, C011-2-1), XO (A002-1-1), ADA (A048-2-1), T-SOD (A001-1-1), CAT (A007-1-1), malondialdehyde (MDA, A003-1-2), interleukin-1β (IL-1β, H002), interleukin-6 (IL-6, H007-1-1), tumor necrosis factor-α (TNF-α, H052-1), and transforming growth factor-β(TGF-β, H034-1) were all bought from Nanjing Jiancheng Bioengineering Institute (Nanjing, China).

### 2.2. Animals

A total of 70 Kunming mice (28 ± 2 g, male, SPF grade) were purchased from Sja Laboratory Animal Co. [Hunan, China, License number: SYXK (Xiang) 2021-0011]. The animal experiment was approved by the Ethics Committee of Laboratory Animals of Nanchang University [SYXK (Gan) 2021-0004]. All mice were housed in individually ventilated cage (IVC) racks under a 12 h light/dark cycle (temperature: 22–25 °C; humidity: 50–60%). The mice were allowed to obtain a standard diet (purchased from Corues Biotechnology, Nanjing, China, catalog number: FZ1252) and water freely. The mice were randomly divided into 7 groups: normal group (Normal), diseased group (Diseased), allopurinol group (Alp), hesperitin group (Hsp), low-dose Hsp-Cu(II) complex group (LHC), medium-dose Hsp-Cu(II) complex group (MHC), and high-dose Hsp-Cu(II) complex group (HHC). Mice except the Normal group were administered intragastrically with oteracil potassium [200 mg/kg body weight (BW)] and hypoxanthine (200 mg/kg) for 2 weeks to establish the HUA diseased mice [34]. Then Alp group mice were given Alp (10 mg/kg), Hsp group mice were given Hsp (60 mg/kg), and the LHC, MHC, and HHC groups mice were given 30, 60, and 120 mg/kg Hsp-Cu(II) complex for 22 days, respectively. In contrast, the mice in the Normal and Diseased groups were only given 0.5% carboxymethyl cellulose sodium (CMC-Na) aqueous solution. All drugs are dissolved in CMC-Na. The BW of each mouse was measured on days 1, 17, and 40. Then 1 h after the final intragastric administration, the rats were killed after intraperitoneal injection of 180 mg/kg of pentobarbital sodium. The whole blood, liver, and kidney were collected, and the organs were weighed. The organ index was calculated based on a previous report [34]. The blood of the mice was centrifuged at 6000× *g* for 15 min to collect the supernatant and stored at −80 °C. One part of the kidneys was soaked in 4% paraformaldehyde for kidney tissue observation. The rest of the kidney and liver were stored at −80 °C for follow-up experiments [34].

### 2.3. Biochemical Assays

The serum was thawed and centrifuged at 12,000× *g* at 4 °C for 15 min. The levels of serum UA and Cr were assayed by a Varioshkan LUX microplate reader (Thermo Scientific, Waltham, MA, USA) and the level of serum BUN was determined by a UV-2450 ultraviolet–visible spectrophotometer (Shimadzu Co., Kyoto, Japan) following the instructions of the test kits.

The supernatants of the liver tissues were taken after homogenized and centrifuged at 12,000× *g* at 4 °C for 15 min. Subsequently, the activity of XO was assayed by the UV-2450 ultraviolet–visible spectrophotometer and the activity of ADA was assayed by the Varioshkan LUX microplate reader according to the description of the test kits.

In addition to the reduced glutathione (GSH)/oxidized glutathione (GSSG) ratio, the activities of total superoxide dismutase (T-SOD) and catalase (CAT), and the content of malondialdehyde (MDA) are also commonly used in certain research areas as important indicators for evaluating oxidative stress. The supernatants of the kidney tissues were taken after homogenized in 10% saline and centrifuged at 12,000× *g* at 4 °C for 15 min. The activities of T-SOD and CAT and the content of MDA were assayed according to the instructions of the kits.

The supernatants of the kidney tissues were taken after homogenized in 10% PBS (pH 7.4) and centrifuged at 12,000× *g* at 4 °C for 15 min. The levels of the four cytokines (IL-1β, IL-6, TGF-β, and TNF-α) were determined by the ELISA method according to the instructions of the kits.

### 2.4. Histopathological Observation

The kidneys of mice soaked in 4% paraformaldehyde were removed and dehydrated sequentially with different concentrations of ethanol. Subsequently, the kidneys were soaked in xylene to replace the ethanol in the tissue and make it transparent. The dehydrated and transparent tissues were immersed in melted paraffin, and the solidified wax was cut into thin slices with a thickness of 5–8 μm on a tissue slicer. After dewaxing the sections, they were stained with hematoxylin and eosin (H&E) solution for 8 min. Finally, the glomerular atrophy and necrosis, and the tubular damage of the renal pathology of each group were observed under an EVOS FL microscope (Thermo Scientific, Waltham, MA, USA) [35].

### 2.5. RT-PCR

Total RNA was extracted from the kidneys using the MiniBEST Universal RNA Extraction Kit (Takara Biomedical Technology Co., Ltd., Beijing, China). The RNA was quantified and then reverse transcribed into cDNA with a PrimeScriptTM RT reagent Kit (Takara Biomedical Technology Co., Ltd., Beijing, China). The RT-PCR was performed using TB Green^®^ Premix Ex TaqTM II (Takara Biomedical Technology Co., Ltd., Beijing, China) with the CFX96 Real-Time PCR Detection System (Bio-Rad Laboratories, Inc., Shanghai, China) to obtain the corresponding mRNA expression level of URAT1, GLUT9, and OAT1. The RT-PCR conditions were as follows: 95 °C for 30 s, then 40 cycles of 95 °C for 5 s, and 60 °C for 30 s. The mRNA expression level was calculated by a method of 2^−ΔΔCt^, and all target gene results were normalized with glyceraldehyde-3-phosphate dehydrogenase (GAPDH) [36]. The primers were synthesized by Sangon Biotech Co., Ltd. (Shanghai, China), and the forward primers and reverse primers were listed in Table 1.

### 2.6. Western Blot

The supernatant of kidney tissues was collected after homogenized in RIPA lysate (Beyotime Biotech. Inc., Shanghai, China), and centrifuged at 10,000× *g* for 10 min. Then a Bicinchoninic acid (BCA) protein assay kit (Beyotime Biotech. Inc., Shanghai, China) was used to determine the total protein content. 8% sodium dodecyl sulfate-polyacrylamide gel electrophoresis (SDS-PAGE) was used to separate protein (20 μg), then the protein was transferred to a polyvinylidene fluoride (PVDF) membrane which soaked in 5% skim milk and blocked for 2 h on Mini Trans-Blot^®^ Cell (Bio-Rad Laboratories, Inc., USA). Then the PVDF membrane was soaked in a primary antibody (Proteintech Group, lnc., Wuhan, China) solution for 12 h. Next, the PVDF membrane was incubated with horseradish peroxidase (HRP) which tagged secondary antibody (Proteintech Group, lnc. Wuhan, China) for 2 h. Finally, after the PVDF membrane was incubated with an ELC kit (Vazyme Biotech Co., Ltd., Nanjing, China), the bands were observed on the ChemiDoc imaging system (Bio-Rad Laboratories, Inc., USA). All results were normalized with GAPDH [37].

### 2.7. Statistical Analysis

Results are expressed as mean ± standard error of mean. The differences were analyzed by a one-way ANOVA followed by Tukey’s post hoc test using SPSS Statistics 22 software (IBM, Armonk, NY, USA). The values were regarded as statistically significant at *p* < 0.05.

## 3. Results

### 3.1. Organ Index and BW of Mice

The difference in body weight (BW) reflects the growth and development of mice [10]. The mortality of the mice was 5%. After the establishment of the HUA-diseased mice on day 17, the BW of normal mice was markedly heavier than the HUA-diseased mice (*p* < 0.05, Figure 1A), indicating that HUA affected the normal growth and development of mice. On day 40, the BW in the LHC, MHC, and HHC groups was greater than that in the Diseased group (*p* < 0.05), but there was little change in Hsp group (*p* > 0.05), indicating that Hsp-Cu(II) complex could restore the HUA diseased mice’s growth and development, and the effect was better than Hsp. The organ index reflects the degree of organ injury, and the higher the organ index, the stronger the organ injury [9]. It was evident that both liver and kidney indexes were greater in the Diseased group than the Normal group. (Figure 1B,C), indicating that the kidneys and liver of the HUA mice were damaged to some extent. The Hsp-Cu(II) complex declined the liver index in HUA mice (*p* < 0.05), but the effect of low-, medium-, and high-doses complex on the liver index was the same, while both Alp and Hsp did not affect the liver index of HUA mice (*p* > 0.05), demonstrating that the complex minimized liver damage to some extent. Additionally, compared to Alp and Hsp, the complex notably decreased the renal index of HUA mice, indicating that the complex reduced renal injury better than Alp and Hsp. Meanwhile, the complex was more effective in protecting the kidneys than the liver in mice.

### 3.2. Serum Physicochemical Indexes of HUA Mice

The main symptom of HUA is increased serum UA level; HUA impairs the excretory function of the kidneys, causing a sharp rise in BUN and Cr levels [35]. As depicted in Figure 2 and Table 2, in the Diseased group, the level of serum UA increased rapidly and it could be seen that the serum UA levels were nearly twice as high as in the Normal group (*p* < 0.05), indicating the establishment of a mouse diseased of hyperuricemia [2]. The serum UA levels in the LHC, MHC, and HHC groups were 1.4, 1.2, and 1.1 times higher than that in the Normal group, respectively, the Alp group was 1.2 times and the Hsp group was 1.5 times. The reducing effect of the medium-dose complex on serum UA was similar to that of Alp (*p* > 0.05), but was markedly superior to the same dose of Hsp (*p* < 0.05), while the reducing effect of high-dose Hsp-Cu(II) complex on UA was better.

Furthermore, in the Diseased group, the serum Cr and BUN levels were abnormally high, 1.5 and 1.9 times greater than those in the Normal group, respectively, demonstrating that the excretory capacity of the kidneys in HUA mice was severely reduced. While in the MHC group, the serum Cr level was decreased by 15.8%, which was only 1.2 times that of the Normal group, while the BUN level was decreased by 40.5% (*p* > 0.05). Hsp also showed some reduction effect in Cr and BUN level, but the effect was only comparable to that of the low-dose complex, with a reduction rate of 9.9% and 26.5%, respectively. Alp reduced Cr and BUN level by only 6.9% and 8.5%, respectively. The above results indicated that Hsp also alleviated kidney injury in the HUA mice, but the protective effect of the Hsp-Cu(II) complex was significantly reinforced, while Alp only showed a weaker protective effect on the kidneys. Luo et al. [38] also found that the renal protective effect of allopurinol was not as good as that of active ingredients from *Moringa Oleifera Lam.* leaves.

### 3.3. Histopathological Observation of the Kidneys in HUA Mice

Excessive levels of UA can form urate deposits in the kidneys, causing serious injury to the kidney tissue [39]. The H&E staining section of the kidney can directly reflect the degree of renal injury. The renal glomeruli of mice in the Normal group (Figure 3A) were in a normal state of the ground, without atrophy and necrosis, and the structure of renal tubules was normal, without edema and fibrosis. In the Diseased group (Figure 3B), there was glomerular atrophy and necrosis, and obvious edema and fibrosis of renal tubules. The mice in the Alp group (Figure 3C) also had glomerular atrophy and necrosis, and renal tubular edema, but the renal fibrosis was well relieved. The morphological characteristics of the kidneys in the Hsp group (Figure 3D) were similar to those of the Diseased group. Notably, in the HUA mice treated with different doses of the Hsp-Cu(II) complex, the symptoms of renal tubular edema were significantly alleviated, with intact glomeruli (Figure 3G). In specific, the morphology of the kidney in the HHC group was proximate to that of the Normal group. Clarifying that the Hsp-Cu(II) complex could restore the damaged kidney tissues in the HUA mice better than ligand Hsp, which was in line with the changes in the renal index. Like Hsp, caffeoylquinic acid derivatives and phloretin had limited protective effects on renal injury, indicating that the formation of the complex with metal ions is an effective way for flavonoids to improve their biological activity [40,41].

### 3.4. Activities of XO and ADA

As the key enzymes for UA synthesis in purine metabolism, XO and ADA have a great influence on the level of serum UA [42]. Figure 1C showed that the liver indexes were greater in the Diseased group than in the Normal group, suggesting that the liver of the HUA mice was damaged to some extent. The treatment of the Hsp-Cu(II) complex could minimize liver damage. Therefore, whether the activities of adenosine deaminase and xanthine oxidase in the livers of each group of mice were affected by the Hsp-Cu(II) complex was determined. As shown in Figure 4A and Table 3, different doses of the Hsp-Cu(II) complex showed significant inhibitory effects on XO activity in the liver. The inhibition rate of XO by high-dose Hsp-Cu(II) complex was 38.4% and by the medium-dose complex also reached 34.0%, which was close to that of Alp. However, the inhibition rate of XO by Hsp was only 10.9%. This was consistent with previous studies that the Hsp-Cu(II) complex exerted a significantly stronger inhibitory effect on XO in vitro than the ligand Hsp [33]. For the inhibition of ADA activity in the liver (Figure 4B), the Normal, Diseased, All, Hsp, and LHC groups exhibited fragile inhibitory ability, with no significant differences, but the MHC and HHC groups significantly inhibited ADA activity with inhibition rates of 18.4% and 23.2% (*p* < 0.05), respectively. It was reported that the inhibitory effect of flavonoids on the activities of XO and ADA was generally lower than those of Alp [43]. After complexing with Cu^2+^, the Hsp-Cu(II) complex strongly inhibited XO and ADA activities, and better inhibited the synthesis of UA in the liver, which was one of the key reasons for the stronger UA-lowering ability of the complex than the ligand (Hsp) [44].

### 3.5. Protein and mRNA Expression Level of UA Transporters in the Kidneys

Next, Western blot and RT-PCR were conducted to analyze the effect of the Hsp-Cu(II) complex on the expression level of three UA transporter proteins (URAT1, GLUT9 and OAT1) in the kidneys [39]. As depicted in Figure 5B–D, the URAT1 and GLUT9 protein levels increased (*p* < 0.05), OAT1 protein level decreased (*p* < 0.05) in the Diseased group compared with the Normal group. The Hsp-Cu(II) complex regulated the URAT1 and GLUT9 expression levels downward and the OAT1 expression level upward. No remarkable differences in the expression of the three transporters among the MHC, HHC, and Normal groups (*p* > 0.05). The effects of Hsp and Alp on the level of the three proteins were similar. In the Hsp and Alp groups, the GLUT9 expression level was close to that in the MHC group (*p* > 0.05), and the URAT1 and OAT1 expression levels were similar to those in the LHC group (*p* > 0.05). Furthermore, the mRNA expression level of the UA transporters in each group of mice was similar to the changes in the protein expression level (Figure 5E–G).

### 3.6. Oxidative Stress in the Kidneys

ROS not promptly scavenged by glutathione, SOD and CAT, as DAMPs, attack mitochondria in the podocyte cells, causing abnormal protease activity and oxidation of DNA, which in turn induces the activation of NLRP3 inflammatory bodies [45]. Therefore, the oxidative stress in the kidney was usually evaluated by the T-SOD and CAT activities and the MDA content [46]. As depicted in Figure 6, in the Diseased group, the T-SOD and CAT activities were notably reduced, while the MDA content was almost twice as high as that of the Normal group, suggesting that a high level of UA could cause serious oxidative stress in the kidneys of mice. Different doses of the complex significantly increased the CAT activity (*p* < 0.05), and the MHC group increased the CAT activity by 27.8%, followed by Hsp (12.8%). Lower effect on T-SOD activity than CAT by the Hsp-Cu(II) complex. The MHC group increased the SOD activity by 6.1%, distinctly different from that in the Diseased group (*p* < 0.05). The MHC group decreased the level of MDA by 31.4%, followed by Hsp (13.1%). Suggesting that both Hsp and Hsp-Cu(II) complex increased SOD and CAT activities to reduce renal oxidative stress, and the effect of the complex was better than Hsp.

### 3.7. NLRP3 Inflammatory Pathway

The NLRP3 pathway contains three proteins and they do not cause inflammation when these three proteins are dissociated in cells [47]. Only when sensing pathogen-associated molecules or danger-associated molecules, the three proteins will gather together to assemble NLRP3 inflammatory bodies and induce inflammation, the ASC (apoptosis-associated speck-like protein containing a CARD) in the nucleus will be released into the cytoplasm as a protein bridge recognition adapter, and its N-terminal PYD will bind to the NLRP3 through oligomerization [48]. Meanwhile, the CARD at the other end recruited Pro-Caspase-1, the precursor of Caspase-1 (cysteinyl aspartate specific proteinase-1), to assemble inactive NLRP3 inflammatory body precursors [15,49]. When the Pro-Caspase-1 on the precursor of the inflammatory body is cut into Caspase-1, it can be activated into bioactive mature NLRP3 inflammatory bodies to mediate the production of a large number of cytokines [50]. In Figure 7, in the Diseased group, the NLRP3, ASC and Caspase-1 protein expression levels were notably raised, (*p* < 0.05). With increasing the dose of the Hsp-Cu(II) complex, the level of NLRP3, Caspase-1, and ASC significantly reduced, and their reductions by the MHC group were 44.2%, 30.0%, and 33.6%, respectively. Interestingly, the three proteins expression levels in the HHC group were found to be close to those in the Normal group. The Alp and Hsp groups showed almost the same effect on the reduction in the three proteins, but their effects were only similar to those of the LHC group (*p* > 0.05), and the reductions were about 33.1% (NLRP3), 25.3% (Caspase-1) and 25.1% (ASC). The aforementioned results clarified that the Hsp-Cu(II) complex better inhibited the NLRP3 inflammatory pathway than Hsp and Alp.

### 3.8. Content of Cytokines in the Kidneys

Inflammatory factors lead to inflammation, which will result in kidney function damage and tissue damage [51]. The Caspase-1 on NLRP3 inflammatory body can activate IL-1β and IL-18, which will bind to MyD88 receptors on the target cells, activating another inflammatory pathway, the NF-κB pathway, which makes a large number of IL-1 and TNF-α transcribed in the cells. These cytokines exacerbate inflammation in the kidneys, causing more severe kidney damage [15,52].

As shown in Figure 8, the four cytokines’ levels were markedly raised in the Diseased group (*p* < 0.05), suggesting that there was severe inflammation and renal fibrosis in HUA mice’s kidney. After being administered intragastrically with the Hsp-Cu(II) complex, the level of cytokines in the MHC group decreased by 30.1% (IL-1β), 28.6% (IL-6), 36.2% (TGF-β), and 24.2% (TNF-α), respectively. In particular, the IL-6 and TNF-α levels in the HHC group showed a dramatic decline (*p* > 0.05). Hsp reduced the four cytokines, but the effect was only the same as that of the LHC group (*p* > 0.05). Alp had the weakest effect, only reducing TGF-β and TNF-α levels by 15.5% and 8.3%, respectively, but there is no discernible effect on IL-1β and IL-6 (*p* > 0.05). Demonstrating that the Hsp-Cu(II) complex exerted a greater inhibitory effect on inflammation than ligand Hsp and Alp, which was consistent with the results of renal histopathological observation. It was also reported that the metal complexes of baicalin with La(III), Y(III), and Nd(III) reduced the TNF-α, IL-6β, and IL-6 contents, and their effects were stronger than that of ligand baicalin [53].

## 4. Discussion

Prolonged hyperuricemia causes uric acid to be deposited throughout the body to form urate, which can be very harmful to the human body, including kidney inflammation and subsequent fibrosis [54]. To date, clinical drugs often cause adverse reactions. Our previous research demonstrated that the Hsp-Cu(II) complex was non-toxic and possessed a significantly higher inhibition ability against xanthine oxidase in vitro than the ligand Hsp [33]. However, the effect of uric acid-lowering by the complex in the body was unclear. In this study, we used potassium oxalate combined with hypoxanthine to treat mice and found that the serum UA level of mice was markedly increased and the kidneys damaged, this indicated the HUA Diseased mice successfully created [55]. Interestingly, the Hsp-Cu(II) complex showed a better uric acid lowering activity in vivo than many natural plant active ingredients. For example, anthocyanin extracted from purple sweet potato and flavonoid extracted from saffron had only 65.3% and 56.7% of the UA-lowering effect compared with Alp in vivo, respectively [36,56], while the reducing effect of the medium-dose Hsp-Cu(II) complex on serum UA was close to that of Alp, and the high-dose complex was better than that of Alp. At the same time, the complex returned the kidney tissue damage to a normal state in hyperuricemia mice, confirming the UA-lowering ability of the Hsp-Cu(II) complex.

The increases in XO and ADA activities will cause the catabolism of purines, and the subsequent rise in the production of uric acid, resulting in a higher blood level of uric acid [57]. Therefore, lowering the activity of XO and ADA can reduce the formation of uric acid, superoxide radicals, and ammonia peroxide in the body and improve the symptoms of hyperuricemia. Such as kaempferol-3′-sulfonate, and chicory aqueous extract inhibited XO activity to reduce serum uric acid levels in HUA mice [58,59]. Similarly, in the present study, the high-dose Hsp-Cu(II) complex significantly inhibited liver XO and ADA activities in HUA mice, equivalent to the positive control allopurinol, which was consistent with the results in vitro [33]. The inhibition of XO by Hsp was weaker than that of the Hsp-Cu(II), and there was no significant inhibition on ADA. Suggesting that the complex decreased the liver XO and ADA activities to reduce uric acid production, which was superior to Hsp.

Moreover, the urate excretion pathway in hyperuricemic mice was further analyzed. The dysregulation of urate transport-associated proteins (e.g., GLUT9, URAT1, OAT1, and OAT3) leads to renal insufficiency with renal uric acid reabsorption increased and excretion decreased, which was supposed to be the reason for the development of hyperuricemia in mice [60]. Our study indicated that the Hsp-Cu(II) complex downregulated URAT1 and GLUT9 protein expression and upregulated OAT1 protein expression. These processes promoted renal uric acid excretion, thereby lowering serum uric acid levels, which were consistent with previous research on reversing GLUT9, URAT1, OAT1, and ABCG2 protein expression levels in the kidneys of rats to reduce the serum uric acid and improve renal tubular dilatation by saponins and emodinol [61,62]. In conclusion, the Hsp-Cu(II) complex reduces uric acid production through two pathways: downregulating XO and ADA activities and modulating the protein expression of urate transporters to promote uric acid excretion.

Notably, hyperuricemia leads to an increase in serum uric acid levels by activating oxidative stress and damaging cells, making tissues hypoxic. Uric acid accumulation may trigger the production of oxidative stress, leading to surging levels of MDA and reduced CAT and SOD activities [63]. In the current study, the renal antioxidant system of the hyperuricemic mice was damaged and showed oxidative stress. The treatment of the middle dose of Hsp-Cu(II) complex increased the CAT activity by 27.8%, increased the SOD activity by 6.1%, and decreased the level of MDA by 31.4%, distinctly different from that in the Diseased group. Suggesting that the Hsp-Cu(II) complex increased SOD and CAT activities, and suppressed the MDA contents to reduce oxidative stress in the kidneys, which was in line with previous reports [64]. Therefore, the Hsp-Cu(II) complex significantly alleviated HUA by improving the activities of antioxidant enzymes, exerting favorable antioxidative effects. Urate deposition and oxidative stress are two important triggers of activation of the NLRP3 inflammatory pathway, which will result in excessive secretion of IL-1β, eventually leading to renal inflammation [48]. This study indicated that the NLRP3 inflammasome was boosted in HUA mice, and the Hsp-Cu(II) treatment prevented this alteration to inhibit the expression of the downstream effectors of the NLRP3 inflammasome (NLRP3, ASC, and caspase-1). Cytokines are major contributors to the inflammatory response in the body and have been reported to cause renal fibrosis, leading to impaired kidney function and tissue damage. The significant reductions of the four cytokines (interleukin-1β, interleukin-6, tumor necrosis factor-α, and transforming growth factor-β) in HUA mice after gavage of the Hsp-Cu(II) complex were observed, pointing that the nephroprotective effect of the complex was partly attributed to the inhibition of inflammatory responses mediated by the activation of NLRP3 inflammasome. These results were consistent with previous studies such as mangiferin, atractylenolide III, and puerarin reduced inflammatory responses by suppressing the NLRP3 inflammasome activation and cytokine expression, alleviated kidney damage [65,66,67]. NLRP3 inflammasomes regulate caspase-1-dependent cellular proptosis and trigger apoptosis under stress, pathological and inflammatory conditions. Chung et al. reported that NLRP3 inflammasomes activated the caspase-8 pathway, forming the NLRP3/ASC/caspase-8 inflammatory complex and inducing apoptosis [68]. Antonopoulos et al. demonstrated that in the absence of caspase-1, NLRP3 inflammasomes directly utilized caspase-8 as both a pro-apoptotic initiator and major IL-1β-converting protease [69]. Lebeaupin et al. suggested that the over-activation of NLRP3 inflammasomes would trigger hepatocyte pyrolysis (caspase-1, -11, interleukin-1β secretion) and apoptosis (caspase-3). Treatment of the inhibitor TUDCA reduced caspase-1, caspase-11, and caspase-3 activities, decreased interleukin-1β secretion, and rescued hepatocyte death [70]. These results suggested that inhibition of inflammasome activation and cell death pathways may be a potential therapeutic approach for disease, but the mechanism of interaction between inflammasome and apoptosis needs to be further explored. Therefore, based on our findings, the potential mechanism of the Hsp-Cu(II) complex attenuated hyperuricemia and renal inflammation may related to inhibiting XO and ADA activities to reduce uric acid synthesis, regulating renal urate transport-related protein expression to promote uric acid excretion, reducing oxidative stress, and inhibiting the activation of NLRP3 inflammasome (Figure 9).

## 5. Conclusions

To conclude, this study indicated the UA-lowering and renal protective functions of the Hsp-Cu(II) complex in HUA mice. The complex showed beneficial effects on the serum levels of UA, creatinine, BUN, inhibition of liver XO and ADA activities, and modulating the urate transporters. In addition, the Hsp-Cu(II) complex significantly suppressed oxidative stress and renal inflammation by blocking the NLRP3 pathway. The Hsp-Cu(II) complex may be a prospective dietary supplementary substance or nutraceutical for the therapy of hyperuricemia. It is desirable to further investigate the intrinsic mechanisms in more detail and the effectiveness of the complex in other HUA diseases.

## Figures and Tables

**Figure 1 foods-13-00591-f001:**
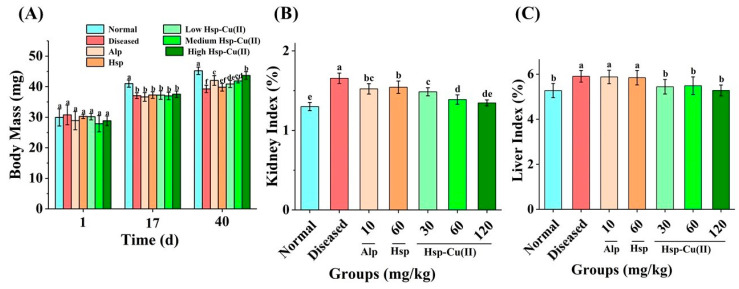
Effect of Hsp-Cu(II) complex on (**A**) body weight, (**B**) kidney index, and (**C**) liver index in the HUA mice. Results are expressed as mean ± SD of 10 samples per group. Values with different superscript letters are significantly different (*p* < 0.05).

**Figure 2 foods-13-00591-f002:**
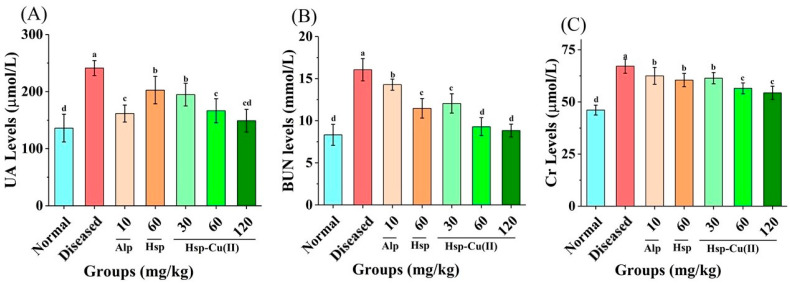
Effect of Hsp-Cu(II) complex on serum physicochemical indexes of (**A**) UA, (**B**) BUN, and (**C**) Cr in the HUA mice. Results are means ± standard deviation, *n* = 10 per group. The different letters represent a significant difference (*p* < 0.05).

**Figure 3 foods-13-00591-f003:**
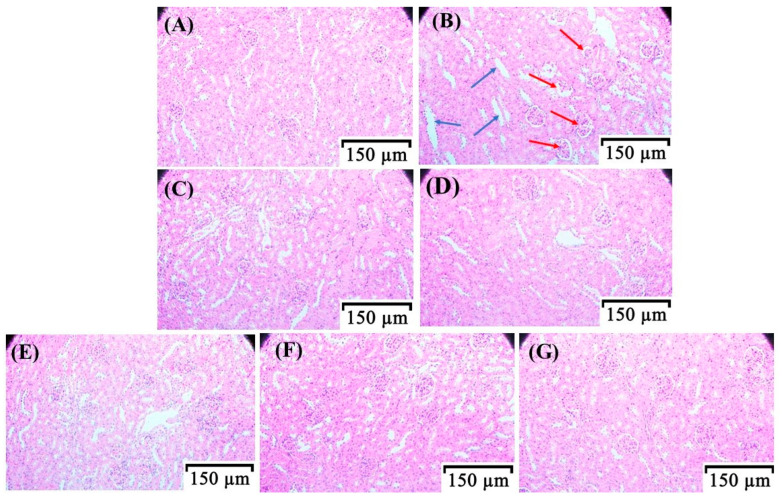
Effect of Hsp-Cu(II) complex on the histopathology of kidney tissue of HUA mice. (**A**) Normal, (**B**) Diseased, (**C**) Alp, (**D**) Hsp, (**E**) LHC, (**F**) MHC, and (**G**) HHC. The red arrows pointed to the glomerular atrophy and necrosis; the blue arrows pointed to the tubular edema and fibrosis.

**Figure 4 foods-13-00591-f004:**
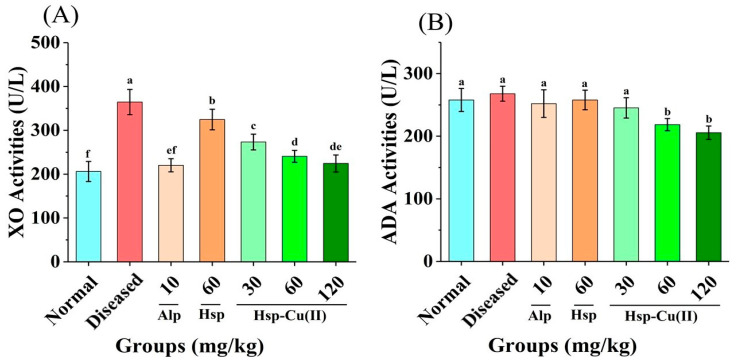
Effect of Hsp-Cu(II) complex on the activity of (**A**) XO and (**B**) ADA in the liver of HUA mice. Results are means ± standard deviation, *n* = 10 per group. The different letters represent a significant difference (*p* < 0.05).

**Figure 5 foods-13-00591-f005:**
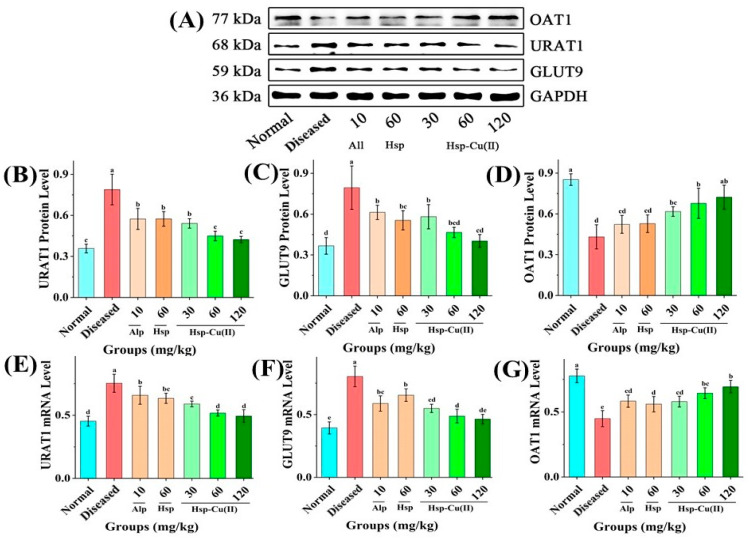
Effect of Hsp-Cu(II) complex on protein mRNA expression of the UA transporters in HUA mice. (**A**) Western blot analysis, the expression level of (**B**) URAT1, (**C**) GLUT9, and (**D**) OAT1. The mRNA expression level of (**E**) URAT1, (**F**) GLUT9, and (**G**) OAT1. Data are presented as the mean ± SD of three independent experiments. The different letters represent a significant difference (*p* < 0.05).

**Figure 6 foods-13-00591-f006:**
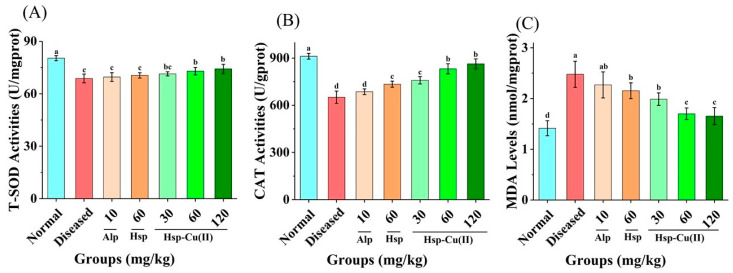
Effect of Hsp-Cu(II) complex on the activities of (**A**) T-SOD, (**B**) CAT, and (**C**) MDA level in HUA mice. Results are means ± standard deviation, *n* = 8 per group. The different letters represent a significant difference (*p* < 0.05).

**Figure 7 foods-13-00591-f007:**
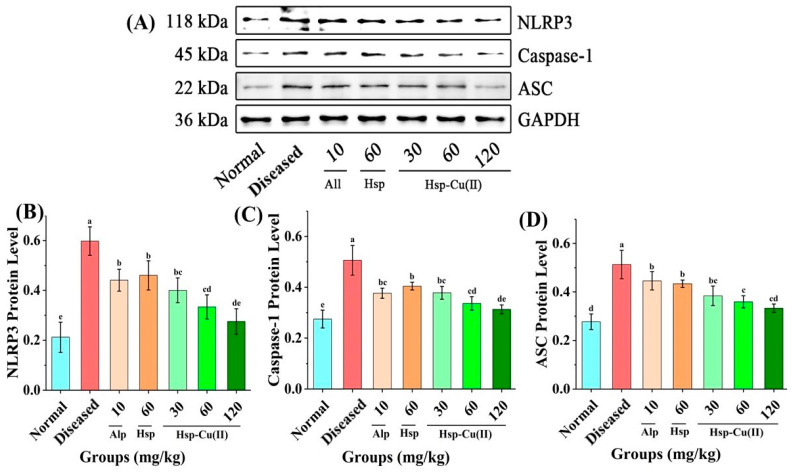
Effect of Hsp-Cu(II) complex on the expression of the proteins involved in NLRP3 inflammatory pathway in the kidneys of HUA mice. (**A**) Western blot analysis, the content of (**B**) NLRP3, (**C**) Caspase-1, and (**D**) ASC. Data are presented as the mean ± SD of three independent experiments. The different letters represent a significant difference (*p* < 0.05).

**Figure 8 foods-13-00591-f008:**
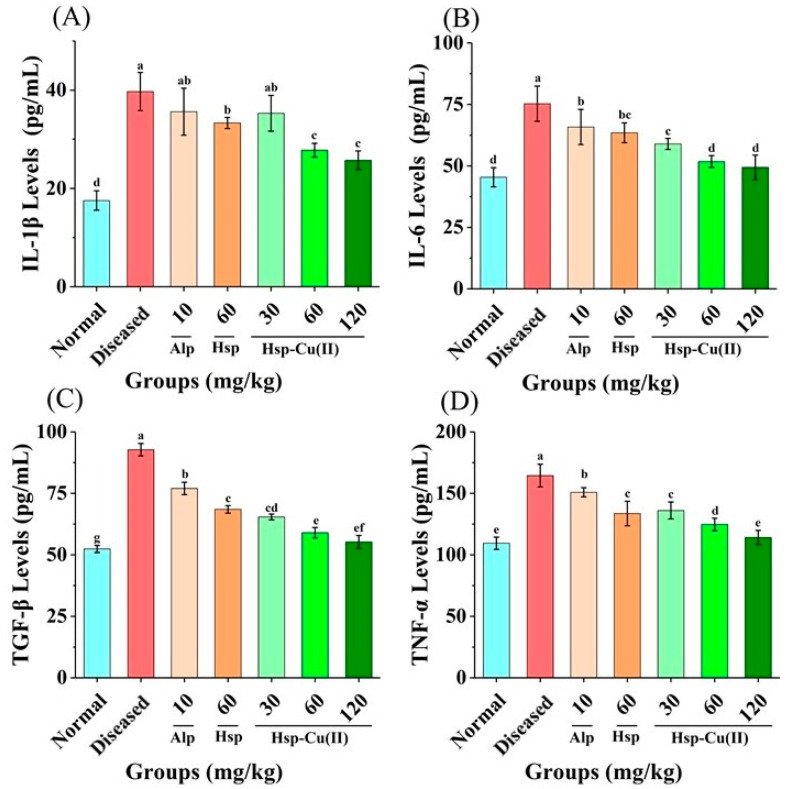
Effect of the Hsp-Cu(II) complex on the level of (**A**) IL-1β, (**B**) IL-6, (**C**) TGF-β, and (**D**) TNF-α cytokines in the kidneys of HUA mice. Results are means ± standard deviation, *n* = 6 per group. The different letters represent a significant difference (*p* < 0.05).

**Figure 9 foods-13-00591-f009:**
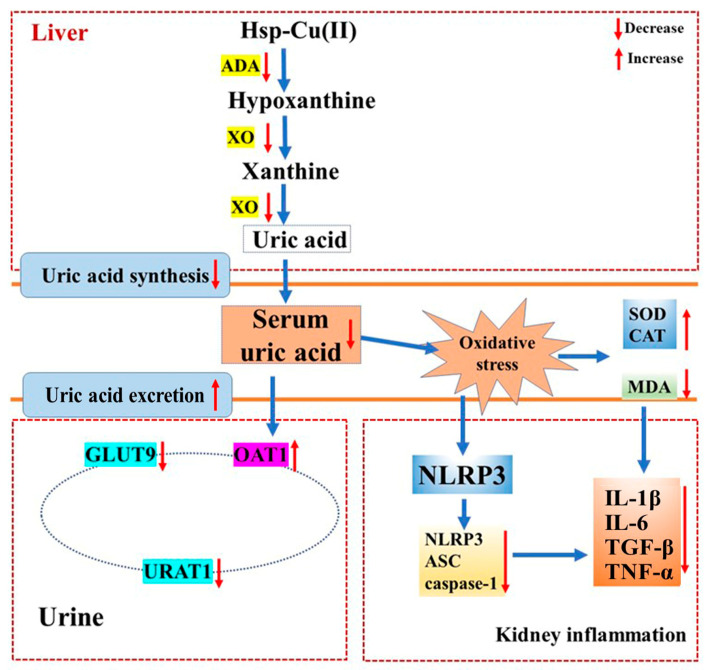
Hsp-Cu(II) complex ameliorates potassium oxalate and hypoxanthine-induced hyperuricemia and kidney inflammation in mice through inhibiting XO and ADA activities and modulating the NLRP3 pathway.

**Table 1 foods-13-00591-t001:** Primer sequences were used in the study. Glucose transporter 9 (GLUT9); glyceraldehyde-3-phosphate dehydrogenase (GAPDH); organic anion transporter 1 (OAT1); urate transporter 1 (URAT1).

Gene	Forward Primer (5′-3′)	Reverse Primer (3′-5′)
URAT1	CAGCACCTTCAGGTCCACAA	TCTGCGCCCAAACCTATCTG
GLUT9	CACCAGCAGAGGAGGACAAA	GCTGGTCTGGAGCACCTCTA
OAT1	GCTGGTACTCCTCCTCTGGA	GGATAGGCATCCATTCCACATT
GAPDH	ACATCATCCCTGCATCCACT	GTCCTCAGTGTAGCCCAAG

**Table 2 foods-13-00591-t002:** The serum physicochemical indexes of mice in each group.

Group	UA (mmol/L)	Cr (µmol/L)	BUN (mmol/L)
Normal	136 ± 24.3 ^d^	46 ± 2.4 ^d^	8 ± 1.2 ^d^
Diseased	241 ± 13.2 ^a^	67 ± 3.4 ^a^	15 ± 1.0 ^a^
Alp	161 ± 14.8 ^b^	62 ± 4.1 ^b^	14 ± 0.7 ^b^
Hsp	202 ± 24.0 ^b^	60 ± 3.3 ^b^	11 ± 1.2 ^c^
LHC	194 ± 19.9 ^b^	61 ± 2.7 ^b^	12 ± 1.1 ^c^
MHC	166 ± 21.1 ^c^	56 ± 2.6 ^c^	9 ± 1.1 ^d^
HHC	149 ± 19.8 ^c^	54 ± 3.2 ^c^	9 ± 0.8 ^d^

Values are means ± standard deviation, *n* = 10 per group. The different letters represent a significant difference (*p* < 0.05). Allopurinol group (Alp); blood urea nitrogen (BUN); creatinine (Cr); hesperitin (Hsp); high-dose Hsp-Cu(II) complex group (HHC); low-dose Hsp-Cu(II) complex group (LHC); medium-dose Hsp-Cu(II) complex group (MHC); diseased group (Diseased); normal group (Normal); uric acid (UA).

**Table 3 foods-13-00591-t003:** The activity of XO and ADA in each group of mice.

Group	XOD Activity (U/L)	Inhibition Rate (%)	ADA Activity (U/L)	Inhibition Rate (%)
Normal	206 ± 23.0 ^f^	-	258 ± 18.5 ^a^	-
Diseased	364 ± 28.9 ^a^	-	268 ± 12.0 ^a^	-
Alp	220 ± 15.0 ^ef^	39.5	252 ± 22.1 ^a^	5.9
Hsp	324 ± 23.5 ^b^	10.9	258 ± 15.7 ^a^	3.7
LHC	273 ± 18.1 ^c^	25.0	245 ± 16.3 ^a^	8.5
MHC	240 ± 13.5 ^d^	34.0	218 ± 9.7 ^b^	18.4
HHC	224 ± 19.4 ^de^	38.4	205 ± 10.8 ^b^	23.2

Values are means ± standard deviation, *n* = 10 per group. The different letters represent a significant difference (*p* < 0.05). Adenosine deaminase (ADA); allopurinol group (Alp); hesperitin (Hsp); high-dose Hsp-Cu(II) complex group (HHC); low-dose Hsp-Cu(II) complex group (LHC); medium-dose Hsp-Cu(II) complex group (MHC); diseased group (Diseased); normal group (Normal); xanthine oxidase (XO).

## Data Availability

Data is contained within the article.

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
