# Peer review of "Hesperitin-Copper(II) Complex Regulates the NLRP3 Pathway and Attenuates Hyperuricemia and Renal Inflammation"

_foods, 2024, doi:10.3390/foods13040591_

Round 1

Reviewer 1 Report

Comments and Suggestions for Authors

Dear all,

Your study is very interesting, however I need some clarifications and I have made some comments and suggestions.

Comments follow throughout the attached document.

Reviewer 2 Report

Comments and Suggestions for Authors

This article describes the effects of Hsp-Cu II complex on uric acid and kidney function. The article is overall well written but minor comments need to be adjusted as:

1. Kindly adjust the abstract to the journal's format and remove the numbering of each section.

2. The authors have used too many abbreviations, so I suggest to compile abbreviations and their meanings at the beginning of the article.

3. In page 3, biochemical assays, what do you mean by g in centrifugated at 12000 x g

4. I suggest changing model group to diseased group or diseased model

5. In page 6, the first paragraph, kindly format Moringa oleifera in italic format and capitalize the genus name, while leaves change to small letters

6. Kindly change figure 3 to a more clear one as it is bluish.

7. why did the authors weigh the liver, while all the work was on the kidney?

1. The authors have used numbers to describe the abstract order and wrote it an structured form. Kindly adjust the abstract to the format of the journal in addition to removing the numbering.

2. I suggest to compile all abbreviations and their respective meaning at the beginning of the article

3. In page 3, biochemical assays, what is meant by g in the centifugation unit used

4. I suggest changing model group to diseased group or diseased model

5. Page 6, the first paragraph, kindly capitalize Moringa as it is a genus name and write the genus 

Comments on the Quality of English Language

Minor editing of English is required

Round 2

Reviewer 1 Report

Comments and Suggestions for Authors

Comments follow throughout the attached document.
